# Replacing Hydrolyzed Soybean Meal with Recombinant β-Glucosidase Enhances Resistance to *Clostridium perfringens* in Broilers Through Immune Modulation

**DOI:** 10.3390/ijms252111700

**Published:** 2024-10-31

**Authors:** Jingxi Huang, Qihang Hou, Ying Yang

**Affiliations:** 1State Key Laboratory of Animal Nutrition and Feeding, College of Animal Science & Technology, China Agricultural University, Beijing 100193, China; huangjx52@163.com; 2College of Animal Science and Technology, Northwest A&F University, Yangling 712100, China; qihanghou1992@163.com

**Keywords:** enzymatic soybean meal, recombinant β-glucosidase, soy isoflavones, *Clostridium perfringens*, immune homeostasis

## Abstract

Aglycone soy isoflavones have notable immune-regulatory bioactivity, while glycosidic forms in soybean meal pose challenges for absorption. β-Glucosidase (EC 3.2.1.21) catalyzes the non-reducing terminal β-d-glucosidic bonds, releasing β-d-glucan and aglycones. This study evaluated the impact of enzymatically hydrolyzed soybean meal (ESM) using recombinant β-glucosidase from *Aspergillus niger* on the growth performance and intestinal immune function of broilers under *Clostridium perfringens* infection. Prior to the feeding trial, soybean meal was enzymatically digested with recombinant β-glucosidase, ensuring almost complete conversion of glycosides to aglycones. After a week of pre-feeding, a total 180 healthy AA broilers were randomly assigned to three groups—control, semi-replacement of ESM (50% ESM), and full-replacement of ESM (100% ESM)—with 6 replicates of 10 chickens, and the trial lasted 28 days. On the 36th day, broilers were challenged with 1 mL of 1 × 10^10^ CFU/mL *Clostridium perfringens* (*Cp*) via gavage for 3 days. The results showed that the substitution of ESM had no effect on the body weight gain of broilers but significantly reduced the feed consumption and feed-to-gain ratio (*p* < 0.01). The study revealed that *Cp* significantly disrupted jejunal morphology, while ESM significantly mitigated this damage (*p* < 0.05). Real-time PCR results demonstrated that compared to the *Cp* group, ESM restored *Cp*-induced intestinal barrier impairments (e.g., *Occludin*, *Claudin-1*, *Muc2*), normalized aberrant cellular proliferation (*PCNA*) and apoptosis (*Caspase-1* and *Caspase-3*), and upregulated the expression of anti-inflammatory factor *Il-10* while suppressing pro-inflammatory cytokines (*Il-1β*, *Il-6*, and *Il-8*) (*p* < 0.05). Moreover, flow cytometry analyses demonstrated that ESM promoted Treg cell-derived Il-10, which alleviated macrophage-derived inflammation. Substituting conventional soybean meal with β-glucosidase, enzymatically treated, significantly reduced feed consumption and alleviated the intestinal damage and immune dysfunctions induced by *Clostridium perfringens* infection in broilers.

## 1. Introduction

Soybean meal, a pivotal protein feedstuff, boasts an array of nutritive compounds, including soy protein, soy isoflavones, soybean saponins, phytosterols, and other bioactive ingredients. Notably, aglycon soy isoflavones, such as daidzein and genistein, have garnered widespread scientific acclaim for their diverse biological benefits, encompassing antioxidant [1], anti-inflammatory [2], cardiovascular prevention [3], anticancer [4], hormone regulatory [5], and other multiple biological functions. Genistein, in particular, has been extensively researched. As a tyrosine kinase inhibitor (TKI) [6], it effectively blocks the activation of transforming growth factor-β-activated kinase 1 (TAK1) and Janus kinase (JAK), subsequently suppressing inflammation-related pathways like NF-κB and STAT [7]. By binding to the active sites of phosphodiesterases (PDEs), genistein decreases their activity, leading to increased intracellular cAMP/cGMP levels, which modulate cellular functions [8]. Its phenolic hydroxyl groups enable free radical scavenging [9], protecting cells from oxidative damage. Furthermore, studies show that genistein modulates intestinal flora composition to improve intestinal homeostasis [10,11]. While these aglycones exhibit heightened biological potency, their implementation in poultry farming faces challenges due to the costly extraction from plants and the impurities inherent in chemical synthesis. Moreover, the predominance of glycosidic isoflavones in soybean meal poses a hurdle as they are biologically inert and challenging for organisms to digest and assimilate [12]. Thus, a pivotal breakthrough lies in facilitating the conversion of glycosidic isoflavones into their more bioactive aglycone forms, significantly enhancing the biological efficacy of soybean meal as a feedstuff.

β-glucosidase (EC 3.2.1.21, BGL), a crucial member of the glycoside hydrolases (GHs) family, hydrolyzes the non-reducing terminal β-d-glucosidic bond, yielding β-d-glucose and the aglycon ligands, and also contributes to cellulose degradation by converting cellobiose into glucose [13]. β-glucosidase has found applications in hydrolyzing various plant glycosides, such as the conversion of polydatins to resveratrol [14], transformation of ginsenosides [15], and hydrolysis of soy isoflavones [16], etc. Plant extraction and microbial fermentation are the main methods employed to acquire β-glucosidase, with the latter favored due to its effectiveness, prompting extensive research efforts. These endeavors encompass enzyme-producing strain screening, mutagenesis of enzyme-producing bacteria, and heterologous expression of enzyme genes [17]. We successfully isolated a GH3 family β-glucosidase from *Aspergillus niger*, achieving a fermentation yield of 70 U/mL in shake flasks, and verified its ability to convert almost all glycosidic soy isoflavones in soybean meal into their aglycone forms through HPLC analysis.

*Clostridium perfringens*, a Gram-positive bacterium ubiquitous in soil, feed, and both healthy and diseased poultry, can cause severe intestinal damage in poultry upon abnormal proliferation and colonization, which lead to intestinal villi necrosis, inflammatory cell infiltration [18], and even mortality [19]. The necrotic enteritis caused by *Clostridium perfringens* incurs annual losses of billions of dollars in the global poultry industry, with the situation being particularly exacerbated following the implementation of restrictions on the use of antibiotic in feed [20]. One study has indicated that genistein can alleviate the detrimental effects of *Clostridium perfringens* on the broiler intestine [21]. Moreover, genistein enhances intestinal barrier function against *Escherichia coli* by modulating barrier integrity, mucosal immunity, and cytokine secretion [22], and alleviates LPS-induced intestinal damage and inflammation in broilers [23]. Genistein alleviates the detrimental effects of harmful bacterial infections by modulating the host’s immune response, ultimately decreasing the reliance on antibiotics within the poultry industry. Consequently, the present experiment aimed to evaluate the impact of enzymatic soybean meal, prepared using recombinant β-glucosidase, on broiler growth performance and intestinal immune homeostasis following *Clostridium perfringens* (*Cp*) infection.

## 2. Results

### 2.1. Content of Soybean Isoflavones in Enzymatic Soybean Meal

The gene encoding β-glucosidase derived from *Aspergillus niger* alongside the heterologously expressed protein (Figure 1A). The enzymatic properties showed that optimal enzyme activity occurs at 40 °C, declining sharply above 50 °C and enzyme activity remains robust within a pH range of 4.5–7, peaking at 5.5 (Figure 1B). Subsequent enzymatic hydrolysis, as evidenced by HPLC (Figure 1C), demonstrates an almost complete conversion of glycosides into aglycon soy isoflavones. Specifically, the conversion rates for daidzin, glycitin, and genistin were 82.31%, 73.84%, and 91.03%, respectively (Table 1). These findings underscore the capability of β-glucosidase sourced from *Aspergillus niger* in transforming glycosidic soy isoflavones present in soybean meal into their aglycon forms.

### 2.2. The Effect of Enzymatic Soybean Meal on Broiler Growth Performance

As shown in Table 2, compared with the control group, the substitution of enzymatic soybean meal at different ratios had no significant effect on the body weight and average daily gain (ADG) of the animals at various stages, whereas it significantly reduced the average daily feed intake (ADFI) and feed conversion ratio (FCR) in the later period and throughout the entire feeding period (*p* < 0.01). These results indicate that enzymatic soybean meal has no impact on the body weight gain of broilers but can reduce feed intake and feed conversion ratio.

### 2.3. Enzymatic SM Ameliorates the Intestinal Epithelial Barrier Function of Broilers Infected with Clostridium perfringens

H&E staining demonstrated that broilers in the *Cp* group exhibited decreased villus height and the ratio of villus height to crypt depth (VH/CD) and enhanced crypt depth compared with broilers in the control group (Figure 2A,B). However, ESM treatment increased the villus height and VH/CD and decreased crypt depth of the jejunum in *Cp*-infected broilers (Figure 2A,B). *Cp* infection results in diminished tight junction protein expression, an effect not significantly reversed by ESM (Figure 2C). *Muc2* expression is significantly upregulated induced by *Cp*, recovered by ESM (*p* < 0.05) (Figure 2D). Collectively, these data demonstrated that ESM ameliorated the intestinal epithelial barrier function of broilers infected with *Clostridium perfringens*.

### 2.4. Enzymatic SM Mitigates Intestinal Epithelial Homeostasis in Jejunum of Broilers Infected with Clostridium perfringens

qRT-PCR analysis explored that genes related to cell proliferation (*PCNA*) and apoptosis (*Caspase-1* and *Caspase-3*) exhibit elevated expression in the *Cp* group, which ESM attenuated (*p* < 0.05) (Figure 3A). Furthermore, expression of *Tlr2*, *Tlr4*, and *Myd88* is reduced in the control and ESM groups compared to the *Cp* group (*p* < 0.05) (Figure 3B). Additionally, *NFκB* and *iNOS* expression is markedly downregulated (*p* < 0.01) in both control and ESM groups compared to the *Cp* group (Figure 3C). These findings suggest that enzymatic soybean meal substitution can alleviate the detrimental effects of *Clostridium perfringens* infection on intestinal epithelial homeostasis.

### 2.5. Enzymatic SM Maintains Immune Homeostasis in Jejunum of Broilers Infected with Clostridium perfringens

Employing flow cytometry, we analyzed the frequency of lamina propria lymphocytes, while qRT-PCR elucidated the expression patterns of pertinent cytokines within jejunal tissues (Figure 4, Figure 5 and Figure 6). Notably, the frequencies of CD45^+^, CD45^+^CD4^+^, and CD4^+^CD25^+^IL-10^+^ cells were significantly elevated (*p* < 0.05) in both the control and ESM groups, in stark contrast to the *Cp* group (Figure 4A). Additionally, *Il-10* mRNA expression demonstrated a highly significant increase (*p* < 0.01) in these groups (Figure 4B), indicating that ESM augments T_reg_-cell activity and IL-10 production in the jejunal lamina propria during *Cp* infection.

Examining macrophage frequencies and cytokine expression, we observed a marked upregulation of macrophages in the *Cp* group (*p* < 0.01) (Figure 5A), which was attenuated by ESM. In parallel, the secretion of pro-inflammatory cytokines (*Il-1β*, *Il-6*, *Il-8*, and *Tnf-α*) by macrophages was significantly diminished (*p* < 0.05) in the control and ESM groups compared to the *Cp* group (Figure 5B). These findings underscore the capacity of ESM to decrease macrophage frequency and expression of pro-inflammatory cytokines triggered by *Cp*.

Further flow cytometric analysis of plasma cell subsets (Bu-1^+^IgY^+^, Bu-1^+^IgA^+^ and Bu-1^+^IgM^+^) in the jejunal lamina propria revealed that *Cp* infection notably augmented the frequency of Bu-1^+^IgA^+^ and Bu-1^+^IgM^+^ plasma cells (*p* < 0.05) (Figure 6). However, the use of ESM demonstrated a mitigating effect on this elevation, highlighting its role in regulating the plasma cell composition in the jejunal lamina propria post-*Cp* challenge.

## 3. Discussion

Soybean meal, a crucial protein source in livestock and poultry diets, typically comprises 20% to 25% of corn–soybean meal-based broiler feeds, which is abundant in soybean isoflavones, and over 95% of the later one exist in glycoside forms. In the whole soybean, isoflavones are stored primarily in the form of glucoside affixes, with β-glucosides (daidzin and genistin) constituting the second largest proportion [24], which are retained as glycosides after heat treatment [25,26] and hydrolyzed to aglycon by β-glucosidase in legume microbial-fermented by-products [27]. Due to their large molecular weight, glycosidic isoflavones have limited intestinal absorption, while once converted to aglycones by β-glucosidase, they are rapidly absorbed within an hour in the duodenum and proximal jejunum [28].

Various β-glucosidases from diverse sources have been employed for isoflavone conversion, including BglAc from thermotolerant *Acidilobus* sp. [29], the cold-adapted gene pgbgl1 cloned from *Psychrobacillus glaciei* sp. [30], and enzyme genes from the *Lactobacillus plantarum* [31], belonging to GH1; and enzymes from the GH3 family of *Aspergillus flavus* origin [32] have proven efficient in transforming glycosidic soy isoflavones into their bioavailable aglycone counterparts. Additionally, soybean-derived glycosidases like GmICHG enhance isoflavone bioavailability [33]. Some researchers are even exploring the possibility of genetic engineering soybeans to directly produce aglycone isoflavones, addressing the fundamental causes [34]. In the previous study, a strain of *Aspergillus niger* with high production of β-glucosidase was obtained, and the enzyme gene was cloned and successfully expressed heterologously, with a molecular weight of 91 KDa, an optimal pH of 5.5, an optimal temperature of 40 °C, and a shaking-flask fermentation yield of 70 U/mL. After enzymatic hydrolysis, the glycosides have been converted almost completely into aglycon form of soybean isoflavones, as can be seen from the HPLC graph, and the conversion rate of daidzin, glycitin, and genistin is 82.3%, 73.8%, and 91.0%, respectively, and this result was similar to that of another study [35].

In soybeans and most soybean-derived feed, the concentration of genistein typically surpasses that of daidzein and glycitein [36], exhibiting enhanced bioavailability [37] and superior biological activity [38,39,40], prompting a predominant focus on genistein in our analysis and a comparison of this experimental outcome with the relevant literature. Our study findings indicate that replacing conventional soybean meal with enzymatic hydrolysis soybean meal (ESM), produced via recombinant β-glucosidase, does not alter the body weight gain of broilers, while significantly reducing feed intake and FCR. The incorporation of recombinant endo-glucanase, either alone or in combination with β-glucosidase, significantly diminishes FCR [41]. This outcome may be attributed to the synergistic action of glucanase and β-glucosidase, where glucanase degrades cellulose into cellobiose, which is further hydrolyzed into glucose by β-glucosidase. This process not only releases nutrients previously entrapped within cellulose but also generates glucose from cellulose degradation, resulting in a notable enhancement in FCR. What is more, one study revealed that β-Glucosidase supplementation in corn–soy diets enhances protein and fat apparent digestibility [42]. In our study, the main focus was on the aglycone forms of soy isoflavones, particularly genistein and daidzein, released after the enzymatic hydrolysis of soybean meal. Studies have demonstrated that supplementing feed with 20–80 mg/kg of genistein can augment the average daily gain of broilers, with a marked reduction in feed conversion ratio (FCR) at 40–80 mg/kg concentrations [43]. As an estrogen, aglycone-type soy isoflavones do not have a direct growth-promoting effect on livestock and poultry; instead, they contribute to enhanced resilience against acute immune challenges through chronic immune modulation. Consequently, to ascertain whether the substitution of enzymatic soybean meal can fortify broilers against sudden pathogenic attacks, our study exposed the birds to *Clostridium perfringens* at 36 days of age.

*Clostridium perfringens*, an opportunistic pathogen that typically resides in the cecum of various organisms, can proliferate excessively in the small intestine, leading to necrotic enteritis, severe deterioration of small intestinal architecture, lamina propria congestion, and infiltration of numerous inflammatory cells [19]. In this study, intestinal morphology analysis revealed that *Cp* challenge inflicted severe damage to villus morphology, marked by extensive necrosis and apical cell shedding, which was mitigated by the use of enzymatic soybean meal (ESM). Prior research concurs that *Cp* invasion significantly disrupts jejunal intestinal morphology in broilers [44]. Notably, ESM did not significantly impact the expression of tight junction proteins in this study, while genistein has demonstrated beneficial effects in other models, such as *Escherichia coli* infection [22] and LPS challenge [45]. Intriguingly, ESM significantly enhanced *Muc2* expression in the mucus layer, a finding echoed in a mouse colitis model [46]. Upon pathogen invasion, the body upregulates genes like inducible nitric oxide synthase (*iNOS*) to mount an immune response [47,48]. Our results showed a marked increase in *iNOS* expression in the *Cp* group, indicative of a robust inflammatory response. The abnormal proliferation and apoptosis of jejunal cells observed in the *Cp* group aligned with the intestinal morphology findings. Further analysis via qRT-PCR revealed that *Cp* invasion upregulated the expression of membrane receptor *Tlr2* and potentially stimulated *Tlr4* expression through its secreted toxins. This activation led to the recruitment of *Myd88*, triggering downstream *NFκB* signaling and regulating immune-related gene expression. Enzymatic hydrolyzed soybean meal partially mitigated this signaling cascade, consistent with previous studies [39,49].

When lymphocytes in the lamina propria of the jejunum encounter antigen–MHC complexes on antigen-presenting cells, they activate helper T-lymphocytes, which subsequently secrete the crucial anti-inflammatory cytokine IL-10 that suppresses pro-inflammatory cytokine secretion while modulating macrophage and T-cell activity [50]. During the initial stages of infection, pro-inflammatory cytokines surge to activate the immune system, inducing macrophages, NK cells, and other immune cells to clear pathogens [51]. In the middle-to-later stages, under the regulation of immune cells and anti-inflammatory cytokines, their expression decreases to prevent tissue damage caused by excessive immune activation. Research indicates that genistein enhances IL-10 levels in splenocyte culture supernatants [52]. The results also showed that the percentage of lymphocytes and T_h_-cells in jejunum in the ESM group was similar to that of the normal group, whereas it was significantly lower in the *Cp* group. Macrophages, vital components of innate immunity, recognize antigens through surface receptors, initiating signaling cascades that present antigens to T cells and release various pro-inflammatory factors (IL-1β, IL-6, IL-8, and TNF-α). It has been shown that genistein can alleviate LPS-induced damage in mice by down-regulating macrophages [53], and that it can attenuate osteoarthritis by reducing IL-6 expression in diabetic mice [54]. The present study also showed the same trend that enzymatically digested soybean meal attenuated macrophage activation and their expression of secreted pro-inflammatory factors after *Cp* infection. Plasma cells, responsible for immunoglobulin secretion, are essential for humoral immunity. While ESM did not significantly impact plasma cells in our study, genistein have been reported to elevate serum IgA and IgG levels [55].

## 4. Materials and Methods

### 4.1. Preparation of Enzymatic Soybean Meal Hydrolyzed by β-Glucosidase (EC 3.2.1.21)

The enzyme employed in this experiment was derived from the fermentation of engineered bacteria. The enzymatic properties are determined according to the provided methods [56]. The protein was dissolved in distilled water to achieve a final concentration of 20 U/mL and mixed with soybean meal (crude protein ≥ 43%, Brazil) at a liquid–solid ratio of 1:1 (*V*/*W*) and then incubated at 37 °C for 12 h before taking samples. Subsequently, 0.5 g of the sample was dissolved in 5 mL of 80% methanol aqueous solution, filtered through a 0.45 μm nylon filter, and subjected to analysis. According to the National Standard of the People’s Republic of China, “Determination of soybean isoflavone in health-care food by high performance liquid chromatography” (GB/T 23788-2009) [57], the content of soybean isoflavones in the enzymatic digestion of soybean meal was determined. The enzymatic soybean meal was dried in an oven at 55 °C.

### 4.2. Experimental Design

The experiment was approved by the Laboratory Animal Welfare and Animal Experimental Ethical Inspection of China Agricultural University (Number: AW02804202-1-1). All efforts were made to minimize animal suffering.

In this study, 180 healthy 1-day-old AA broilers, exhibiting comparable initial body weights, were randomly allocated into three groups after a week-long pre-feeding—control group fed with a basal diet, an enzymatic soybean meal semi-replaced group (50% ESM), and an enzymatic soybean meal full-replaced group (100% ESM—with 6 replicates of 10 chickens in each, and the period was 28 days, during which the chickens had ad libitum access to feed and water. Broilers were reared in enclosed barns and ventilated through exhaust fans. Each replicate is housed in individual cages made of double-layered galvanized wire mesh. After the official start of the experiment, the temperature was gradually reduced from 35 °C to a constant temperature of 25 °C, while humidity was maintained at around 70%. The chickens were provided with a guaranteed 23 h of light and 1 h of darkness each day. The basal diet, a corn–soybean meal blend, adhered to the Chinese broiler feeding standard (2004) and is detailed in Table 3. On the 36th day, 1 mL (1 × 10^10^ CFU/mL) *Clostridium perfringens* (*Cp*) was orally gavaged to the craw after 8 h of fasting for 3 consecutive days. For gavage administration, fresh bacterial suspensions were used; the non-infected control group was gavaged with liquid medium instead. At the end of the gavage, 2 chickens were selected from each replicate for slaughter sampling, with a total of 12 chickens in each group.

### 4.3. Cultivation of Clostridium perfringens

*Clostridium perfringens* (*Cp*) type A strain (CVCC52, China Veterinary Culture Collection Center, Beijing, China) was used for the infection model. The previously preserved *Clostridium perfringens* in the laboratory was streaked onto tryptose–sulfate–cycloserine Agar (TSCA) medium containing d-cycloserine (Aobox, Beijing, China). After anaerobic incubation at 37 °C for 24 h, a single clone was picked and inoculated into mercaptoacetic acid salt liquid medium (Aobox, Beijing, China) for 15 h. The sterile glycerol was added to the bacterial suspension to achieve a final concentration of 25% before being stored at −80 °C. The frozen bacteria were inoculated at a 3% ratio into fresh liquid medium for anaerobic culture for 18 h and subsequently counted on agar medium.

### 4.4. HE Staining of Jejunum

Jejunum mid-section tissues (1 cm) were excised, rinsed with PBS, and put in 4% paraformaldehyde solution. Following paraffin embedding, the tissues were subjected to hematoxylin and eosin (HE) staining. The damaged state of the intestinal tract was observed and photographed under a microscope (Leica DM1000, Wetzlar, Germany), the villus height (VH) and crypt depth (CD) were measured with Image pro plus 6.0 software, and the ratio of VH to CD was calculated.

The trace mineral premix provided the following per kilogram of diets: Mn 100 mg, Zn 75 mg, Fe 80 mg, Cu 8 mg, Se 0.25 mg, and I 0.35 mg. The vitamin premix provided the following per kilogram of diets: VA 9 500 IU, VD3 2 500 IU, VE 30 IU, VK3 2.65 mg, VB1 2 mg, VB6 6 mg, VB12 0.025 mg, biotin 0.032 5 mg, folic acid 1.25 mg, pantothenic acid 12 mg, and nicotinic acid 50 mg. Metabolizable energy and crude protein in the nutrient levels are measured values; the rest are calculated values.

### 4.5. RNA Extraction and Relative Quantitative Real-Time PCR

Total RNA was extracted from jejunal tissue using the Trizol (GeneStar, Beijing, China) method, and mRNA was obtained via a PrimeScript™ RT reagent Kit (Takara, Tokyo, Japan). According to TB Green^®^ Premix Ex Taq™ (Takara, Tokyo, Japan) instructions, mRNA, primers shown in Table 4, and related reagents were added to a 96-well plate, and the relative expression level of mRNA was measured using the Applied Biosystems 7500 real-time PCR system (Thermo Fisher Scientific, Waltham, MA, USA).

### 4.6. Separation of Jejunal LPLs

The jejunum was cut and rinsed with PBS. The jejunum was incubated in D-PBS (5 mM EDTA, 0.154 mg/mL dithiothreitol, 5% FBS, 100 U/mL penicillin and 0.1 mg/mL streptomycin) for 30 min (37 °C, 120 rpm) to remove the epithelial cells. After rinsing twice in PBS, it was clipped into 1 mm pieces. The cells were digested in collagenase solution (1640 medium, 100 U/mL penicillin, 0.1 mg/mL streptomycin, 1% glutamine, 5% FBS, 0.5 mg/mL collagenase/dispersase, and 1 U/mL DNase 1) for 30 min (37 °C, 120 rpm), filtered (40 μm), rinsed twice in PBS, and purified from the epithelial cells using a 40%/80% Percoll gradient (Solarbio, Beijing, China).

### 4.7. Flow Cytometry

Appropriate amounts of surface antibodies (CD45-APC (SouthernBiotech, Homewood, AL, USA, F2100403), CD4-PE/CY7 (SouthernBiotech, 8210-17), CD25-PE (Thermo Fisher Scientific, Waltham, MA, USA, 12-0259-42), IL-10-FITC (Biolegend, Beijing, China, 505007), CD11c-PE/CY7 (SouthernBiotech, 8420), Bu-1-APC (SouthernBiotech, 8310-09), IgA-Pacific Blue (SouthernBiotech, 8395-31), IgM-PE (SouthernBiotech, 8395-09), IgY-FITC (SouthernBiotech, 8320-02) were added, according to the instructions, and incubated for 30 min at 4 °C, protecting from light. A total of 1~2 mL of staining buffer was added and washed once at 4 °C, and the supernatant was discarded via centrifugation at 500× *g* for 5 min. The supernatant was resuspended with 500 μL staining buffer, transferred to a 5 mL flow-through tube, and filtered through a 40 μm filter before loading. Samples were examined with a FACSCanto™ II flow analyzer (BD, Franklin Lakes, NJ, USA) and analyzed with FlowJo v10.6.2 software (TreeStar, Chico, CA, USA), using isotype photos or unstained blank tubes to circle the gate.

### 4.8. Statistics and Analysis

The experimental data were expressed as mean ± standard deviation (SD). SPSS 19.0 software was applied to statistically analyze the experimental data, and the significance of difference was tested using One-Way ANOVA. Graphpad prism 8.0 software was used to graph the data. A statistically significant difference was defined when the *p*-value is less than 0.05.

## 5. Conclusions

Although enzymatic digestion of soybean meal by β-glucosidase did not impact body weight gain, it had a significant effect on feed consumption and FCR, which can achieve cost efficiencies. Furthermore, when suddenly exposed to *Clostridium perfringens*, ESM could mitigate the adverse effects of *Cp* infection by modulating T_reg_ cell/macrophage homeostasis, which, in turn, increases the expression of the anti-inflammatory factor *Il-10* and decreases the expression of pro-inflammatory factors so that enzymatic digestion of soybean meal can replace the exogenous addition of aglycon-type soy isoflavones for immunomodulatory effects.

## Figures and Tables

**Figure 1 ijms-25-11700-f001:**
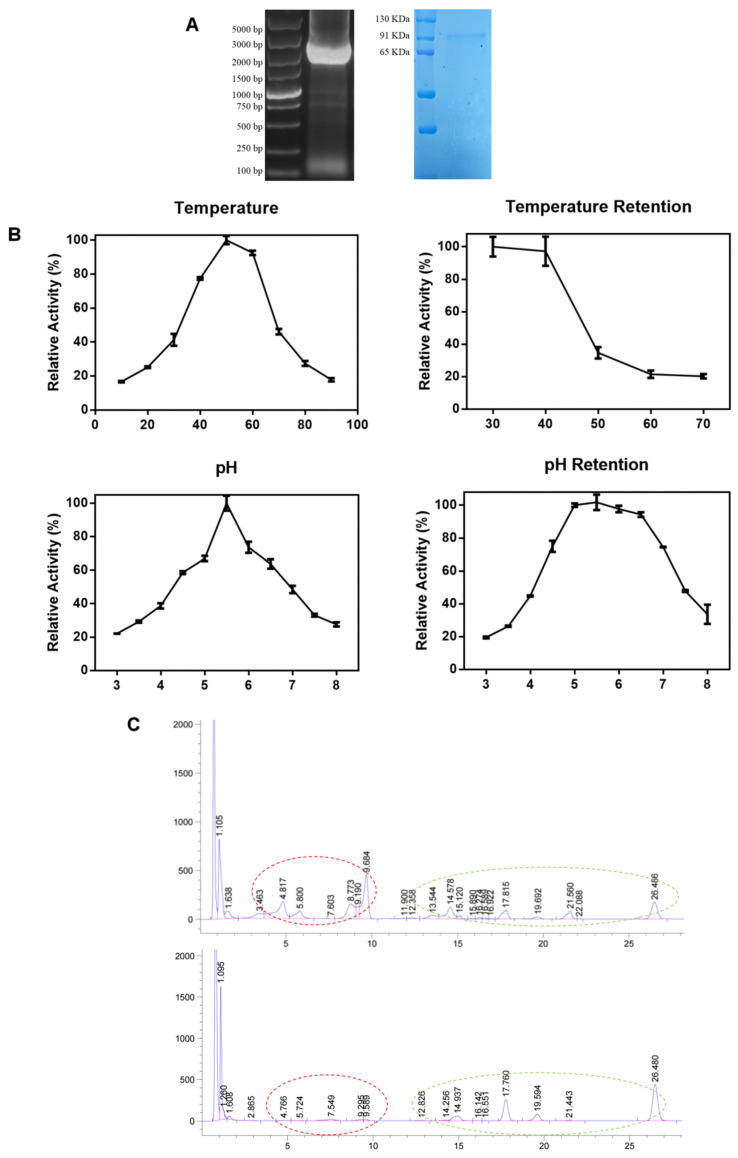
Schematic diagram of enzymatic digestion of soybean meal: (**A**) agarose gel of the β-glucosidase gene originated from *Aspergillus niger* and SDS-PAGE of heterologously expressed protein; (**B**) effects of temperature and pH on the activity and stability of recombinant protein; (**C**) the HPLC results of soybean isoflavones content in soybean meal before and after enzymatic digestion. The red circle indicated the content of glycoside forms of soy isoflavones, while the green circle indicated the content of aglycone forms.

**Figure 2 ijms-25-11700-f002:**
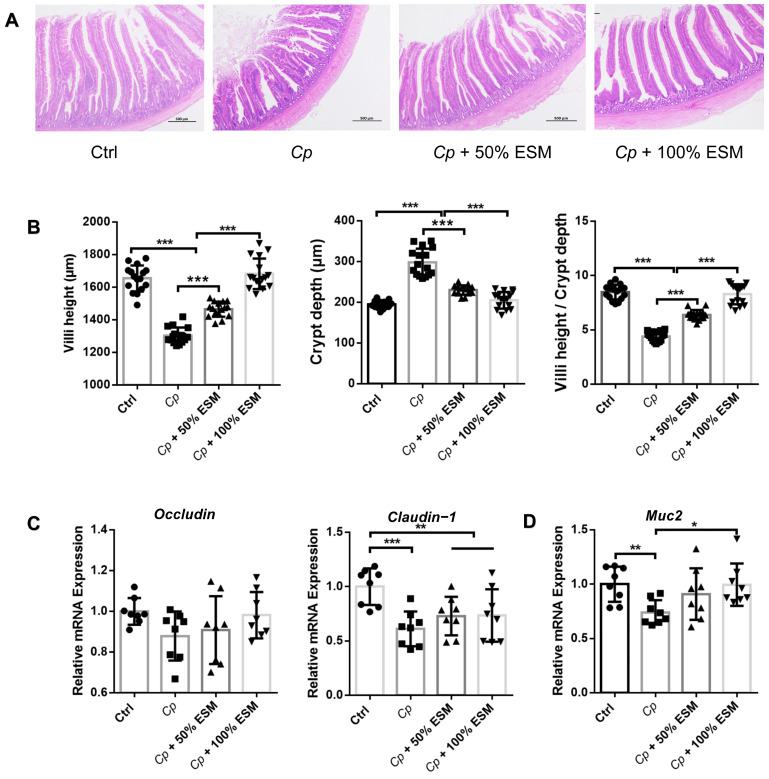
Enzymatic SM ameliorates the intestinal epithelial barrier function of broilers infected with *Clostridium perfringens*: (**A**) HE staining of jejunum (×40 magnification, scale bar = 500 μm); (**B**) the villus height (VH), crypt depth (CD), and the ratio of VH and CD (VH/CD) in duodenum, n = 16; (**C**) qRT-PCR analysis of tight junction, n = 8; (**D**) qRT-PCR analysis of mucin2, n = 8. Ctrl group, a basal diet; *Cp*, basal diet infected with *Cp*; 50% ESM, semi-replacement of enzymatic soybean meal diet infected with *Cp*; 100% ESM, full-replacement of enzymatic soybean meal diet infected with *Cp*. The differences among groups were determined via ANOVA. The results are presented as mean ± SD. * *p* < 0.05, ** *p* < 0.01, *** *p* < 0.001.

**Figure 3 ijms-25-11700-f003:**
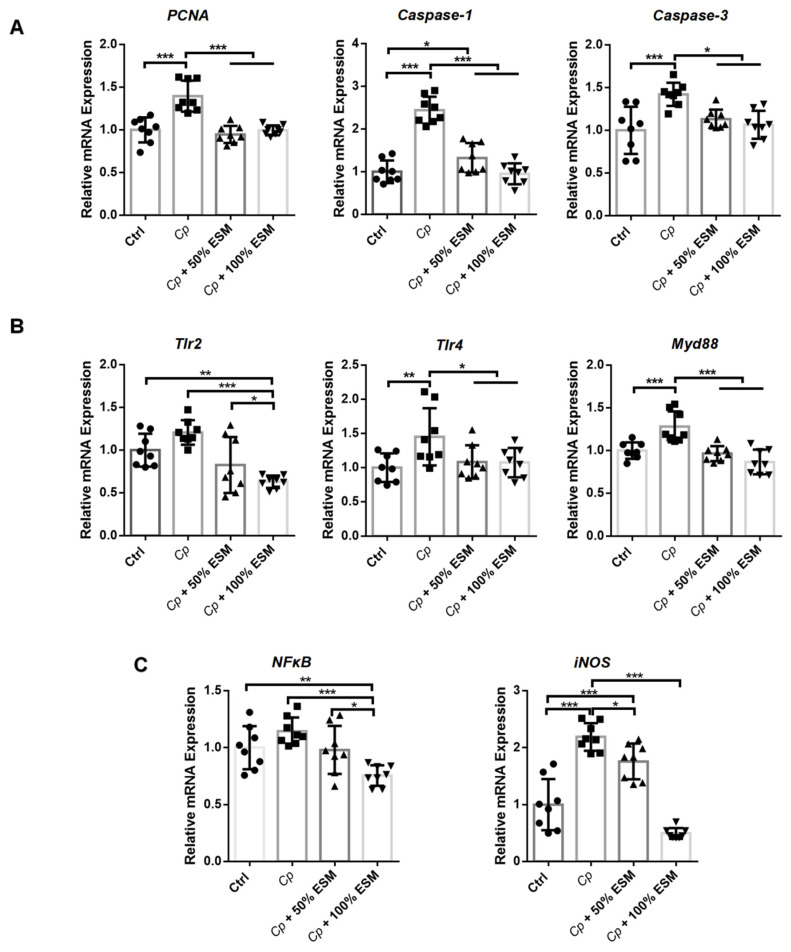
Enzymatic SM mitigates intestinal epithelial homeostasis in jejunum of broilers infected with *Clostridium perfringens*: (**A**) qRT-PCR analysis of cell proliferation and apoptosis genes; (**B**) qRT-PCR analysis of toll-like receptor pathway-related genes; (**C**) qRT-PCR analysis of *NFκB* and *iNOS*. Ctrl group, a basal diet; *Cp*, basal diet infected with *Cp*; 50% ESM, semi-replacement of enzymatic soybean meal diet infected with *Cp*; 100% ESM, full-replacement of enzymatic soybean meal diet infected with *Cp*. The differences among groups were determined via ANOVA. The results are presented as mean ± SD, n = 8. * *p* < 0.05, ** *p* < 0.01, *** *p* < 0.001.

**Figure 4 ijms-25-11700-f004:**
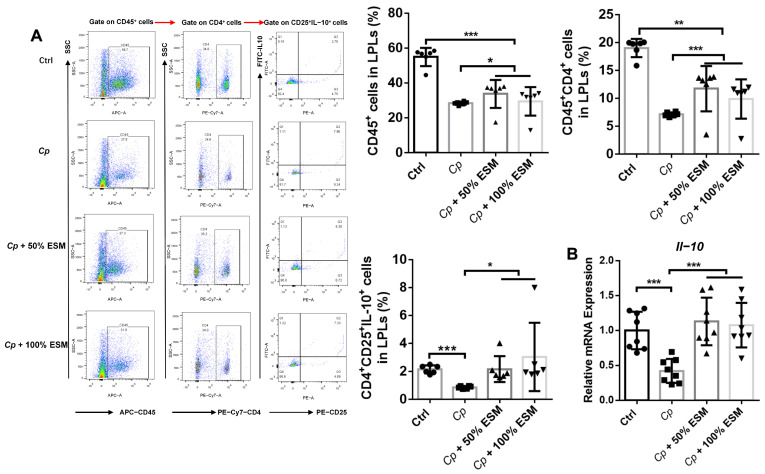
Enzymatic SM promotes the expression of T cell and IL-10 expression in jejunum of broilers infected with *Clostridium perfringens*: (**A**) flow cytometry analysis of CD45^+^, CD45^+^CD4^+^, CD4^+^CD25^+^IL-10^+^ cell frequency in jejunum LPLs of broilers, n = 6; (**B**) qRT-PCR analysis of *Il-10* in jejunum, n = 8. Ctrl group, a basal diet; *Cp*, basal diet infected with *Cp*; 50% ESM, semi-replacement of enzymatic soybean meal diet infected with *Cp*; 100% ESM, full-replacement of enzymatic soybean meal diet infected with *Cp*. The differences among groups were determined via ANOVA. The results are presented as mean ± SD. * *p* < 0.05, ** *p* < 0.01, *** *p* < 0.001.

**Figure 5 ijms-25-11700-f005:**
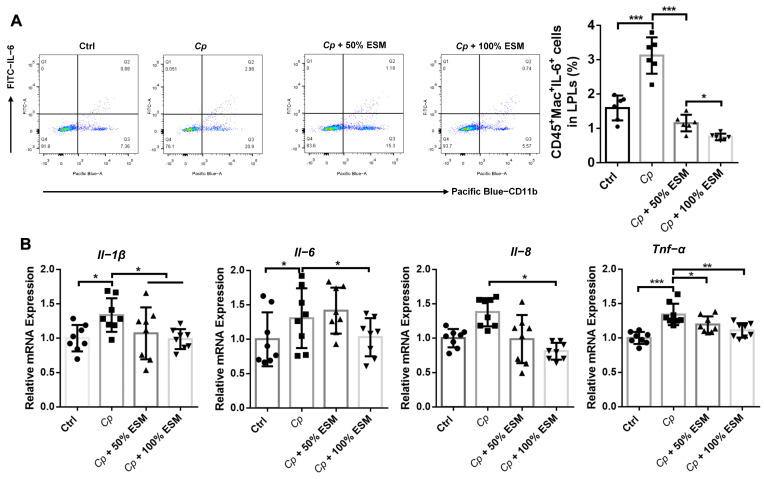
Enzymatic SM alleviates the expression of macrophages and related cytokines in jejunum of broilers infected with *Clostridium perfringens*: (**A**) flow cytometry analysis of CD45^+^Mac^+^IL-6^+^ cell frequency in jejunum LPLs of broilers, n = 6; (**B**) qRT-PCR analysis of cytokines in jejunum, n = 8. Ctrl group, a basal diet; *Cp*, basal diet infected with *Cp*; 50% ESM, semi-replacement of enzymatic soybean meal diet infected with *Cp*; 100% ESM, full-replacement of enzymatic soybean meal diet infected with *Cp*. The differences among groups were determined via ANOVA. The results are presented as mean ± SD. * *p* < 0.05, ** *p* < 0.01, *** *p* < 0.001.

**Figure 6 ijms-25-11700-f006:**
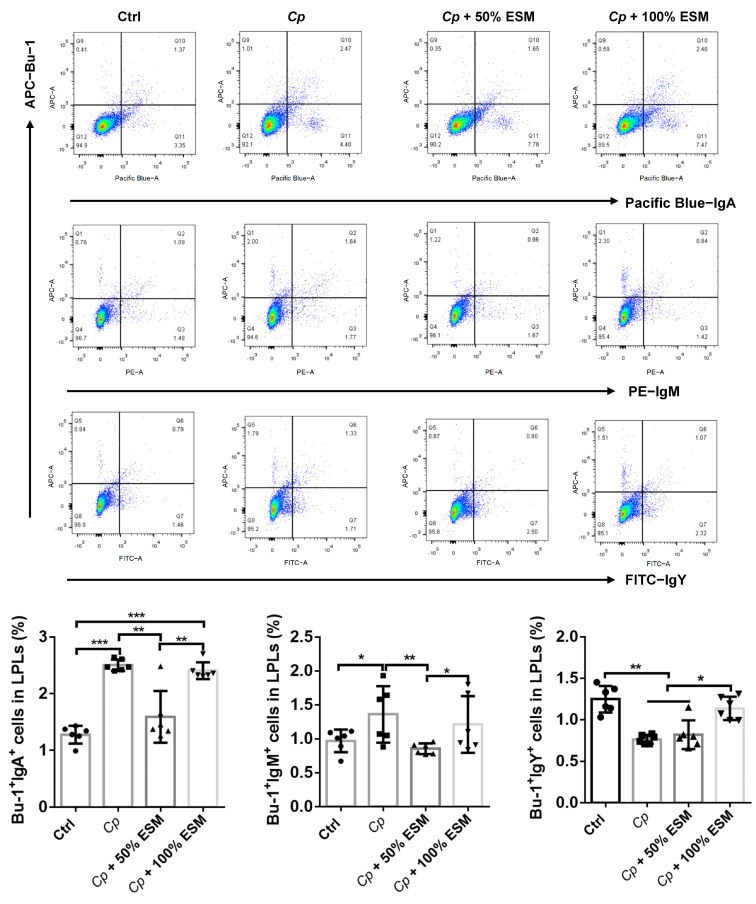
The effects of enzymatic SM on intestinal lamina plasma-cells-mediated humoral immunity of broilers infected with *Clostridium perfringens*. Flow cytometry analysis of Bu-1^+^, Bu-1^+^IgY^+^, Bu-1^+^IgA^+^, and Bu-1^+^IgM^+^ cell frequency in jejunum LPLs of broilers. Ctrl group, a basal diet; *Cp*, basal diet infected with *Cp*; 50% ESM, semi-replacement of enzymatic soybean meal diet infected with *Cp*; 100% ESM, full-replacement of enzymatic soybean meal diet infected with *Cp*. The differences among groups were determined via ANOVA. The results are presented as mean ± SD, n = 6. * *p* < 0.05, ** *p* < 0.01, *** *p* < 0.001.

**Table 1 ijms-25-11700-t001:** The changes of soybean isoflavones content and conversion rate in soybean meal before and after enzymatic digestion (n = 3).

	Before Enzymatic Digestion (mg/kg)	After Enzymatic Digestion (mg/kg)	Conversion Rate (%)
Daidzin	45.06 ± 4.43	0.21 ± 0.00	-
Daidzein	11.89 ± 0.37	48.88 ± 2.15	82.31 ± 3.94
Glycitin	24.82 ± 0.79	0.80 ± 0.04	-
Glycitein	2.77 ± 0.29	21.06 ± 1.68	73.84 ± 7.92
Genistin	58.23 ± 2.95	2.32 ± 0.61	-
Genistein	14.94 ± 0.74	67.94 ± 3.12	91.03 ± 3.43

**Table 2 ijms-25-11700-t002:** The effect of enzymatic SM on the growth performance of broilers.

	Ctrl	50% ESM	100% ESM	*p*-Value
8–21 d				
BW (g)	659.68 ± 15.21	628.68 ± 24.38	630.64 ± 43.23	0.168
ADG (g)	41.19 ± 1.21	38.50 ± 2.21	38.59 ± 3.81	0.114
ADFI (g)	65.11 ± 1.72	63.25 ± 2.35	61.94 ± 2.75	0.090
FCR (g/g)	1.58 ± 0.03	1.65 ± 0.09	1.61 ± 0.09	0.366
22–35 d				
BW (g)	1816.58 ± 65.18	1725.67 ± 47.83	1744.12 ± 74.96	0.052
ADG (g)	91.51 ± 5.16	86.72 ± 3.72	88.16 ± 3.48	0.160
ADFI (g)	157.96 ± 6.12	143.88 ± 3.90 *	144.08 ± 7.45 *	0.043
FCR (g/g)	1.94 ± 0.05	1.87 ± 0.05 **	1.84 ± 0.06 **	0.009
35 d				
ADG (g)	67.51 ± 2.91	63.86 ± 2.03	64.57 ± 3.09	0.077
ADFI (g)	123.49 ± 4.09	114.39 ± 3.34 **	113.87 ± 5.46 **	0.003
FCR (g/g)	1.83 ± 0.03	1.79 ± 0.02 **	1.76 ± 0.03 **	0.004

Ctrl group, a basal diet; 50% ESM, semi-replacement of enzymatic soybean meal diet; 100% ESM, full-replacement of enzymatic soybean meal diet, n = 6. The differences among groups were determined via ANOVA. The results are presented as mean ± SD, n = 6. * *p* < 0.05, ** *p* < 0.01, vs. Ctrl group. Abbreviations: BW, body weight; ADG, average daily gain; ADFI, average daily feed intake; FCR, the ratio of feed to gain.

**Table 3 ijms-25-11700-t003:** Composition and nutrient levels of basal diets (air-dry basis).

Ingredients	7–21 d	22–35 d
Corn	59.63	66.03
Soybean meal (43%)	30.05	24.22
Soybean oil	1.46	1.26
Corn gluten meal	4.69	4.69
Calcium hydrophosphate	1.90	1.57
Limestone	0.91	0.94
NaCl	0.35	0.35
L-lysine HCl (78%)	0.20	0.20
DL-Methionine	0.16	0.09
Choline chloride (50%)	0.30	0.30
Multimineral ^1^	0.20	0.20
Multivitamin ^2^	0.02	0.02
Antioxidants	0.13	0.13
Total	100.00	100.00
Nutrient levels ^3^		
Metabolizable Energy/(MJ/kg)	12.68	12.65
Crude Protein	23.08	20.46
Lys	1.15	1.01
Met	0.50	0.41
Ca	1.00	0.90
AP	0.45	0.40

^1^ The trace mineral premix provided the following per kilogram of diets: Mn 100 mg, Zn 75 mg, Fe 80 mg, Cu 8 mg, Se 0.25 mg, I 0.35 mg. ^2^ The vitamin premix provided the following per kilogram of diets: VA 9 500 IU, VD3 2 500 IU, VE 30 IU, VK3 2.65 mg, VB1 2 mg, VB6 6 mg, VB12 0.025 mg, biotin 0.032 5 mg, folic acid 1.25 mg, pantothenic acid 12 mg, nicotinic acid 50 mg. ^3^ Metabolizable energy and crude protein in the nutrient levels are measured values, the rest are calculated values.

**Table 4 ijms-25-11700-t004:** Primer pairs for qRT-PCR analysis.

Gene	Primer Sequences (5′-3′)	Accession NO.
*β-actin*	F: GAGAAATTGTGCGTGACATCA	NM_205518.1
	R: CCTGAACCTCTCATTGCCA	
*Occludin*	F: AGTTCGACACCGACCTGAAG	NM_205128.1
	R: TCCTGGTATTGAGGGCTGTC	
*Claudin-1*	F: GGTATGGCAACAGAGTGGCT	NM_001013611
	R: CAGCCAATGAAGAGGGCTGA	
*Muc2*	F: TTCATGATGCCTGCTCTTGTG	XM_040673077.1
	R: CCTGAGCCTTGGTACATTCTTGT	
*iNOS*	F: CCTGTACTGAAGGTGGCTATTGG	NM_204961.1
	R: AGGCCTGTGAGAGTGTGCAA	
*PCNA*	F: AATGCGGATACGTTGGCTCT	NM_204170.3
	R: CACCAATGTGGCTGAGGTCT	
*Caspase-1*	F: AGTACGGTGGTGTTCTCCTT	XM_015295935.4
	R: GATCTCATCCGTCATGCTGC	
*Caspase-3*	F: CAGCTGAAGGCTCCTGGTTT	XM_015276122.4
	R: GCCACTCTGCGATTTACACG	
*Tlr-2*	F: CGGTCATCTCAGCTACACCA	NM_204278
	R: GCATCGCATGAAAGACAGGC	
*Tlr-4*	F: GATGCATCCCCAGTCCGTG	NM_001030693
	R: CCAGGGTGGTGTTTGGGATT	
*NFκB*	F: TGGAGAAGGCTATGCAGCTT	NM_205134.1
	R: CATCCTGGACAGCAGTGAGA	
*Myd88*	F: TGCAAGACCATGAAGAACGA	NM_001030962.4
	R: TCACGGCAGCAAGAGAGATT	
*Il-10*	F: CGCTGTCACCGCTTCTTCA	NM_000572.2
	R: TCCCGTTCTCATCCATCTTCTC	
*Il-1β*	F: GCCTGCAGAAGAAGCCTCG	NM_204524.2
	R: GGAAGGTGACGGGCTCAAAA	
*Il-6*	F: CTCCTCGCCAATCTGAAGTC	NM_204628.1
	R: GGCACTGAAACTCCTGGTCT	
*Il-8*	F: TTGCCAAGGAGTGCTAAAGAA	NM_000584.3
	R: GCCCTCTTCAAAAACTTCTCC	
*Tnf-α*	F: GAGCGTTGACTTGGCTGTC	NM_000594.3
	R: AAGCAACAACCAGCTATGCAC	

## Data Availability

Data and materials supporting the findings of this work are available from the corresponding author upon reasonable request.

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
