# Peer review of "Replacing Hydrolyzed Soybean Meal with Recombinant β-Glucosidase Enhances Resistance to Clostridium perfringens in Broilers Through Immune Modulation"

_ijms, 2024, doi:10.3390/ijms252111700_

Round 1
Reviewer 1 Report
Comments and Suggestions for Authors
The replacement of hydrolyzed soybean meal by recombinant β-glucosidase enhances resistance to Clostridium Perfringens in broilers
Dear Authors,
The manuscript is interesting and well prepared. Results confirm positive effect of enzymatic digestion soybean meal by β-glucosidase. Some corrections are required in the text, but mainly involved with text editing and Table 3.
Below I add some suggestions helpful in this process:
In title of manuscript
In case of binominal nomenclature genus Clostridium starts from capital letter, but species name prefringens starts from small letter.
Abstract, Background, 4th line from the bottom
In text of the manuscript is ‘…with recombinant β-glucosidase , ensuring almost complete…’. Space after β-glucosidase must be deleted.
Abstract, Results, end of 2nd line from the top
In text of the manuscript is ‘… and FCR (P<0.01)…’ , but probability was determined on basis on sample of birds from population, in this case p-value is better description.
And other P-values must be also changed to p-value.
Footer, bottom left corner
Current information about year of publication and volume must be added.
Second page, 2nd-3rd line, first reference in text
In text is ‘…encompassing antioxidant[1]’. Normally in entire text of manuscript references are given after space.
Subsection 2.1. Enzymatic soybean meal preparation, 3rd line in first paragraph
Font of reference [15] must be changed.
Third page, Table 1
In case of ingredients, soybean meal crude protein content (%) can be added.
Instead of Lysine, L-lysine HCl (78%) must be used.
ME (metabolizable energy), first slash can be deleted
First two lines description of premix content, font must be adapted to the rest of the text.
Subsection 2.8. Statistics and analysis
Last sentence describing significance levels, font must be adapted to the rest of lines in this paragraph
Table 3
Maybe better will be to add one more row of table under the table and add information about values before and after enzymatic digestion, because now this table is problematic in read. Number of decimals in case of mean value and sd must be the same.
Item |
Before enzymatic digestion |
After enzymatic digestion |
||||
Daidzin |
Glycitin |
Genistin |
Daidzin |
Glycitin |
Genistin |
|
Prezymolysis (mg/kg) |
45.06 ± 4.43 |
24.82 ± 0.79 |
58.23 ± 2.95 |
11.89 ±0.38 |
2.77 ± 0.29 |
14.94 ± 0.74 |
Postzymolysis (mg/kg) |
0.21 ± 0.00 |
0.80 ± 0.04 |
2.32 ± 0.61 |
… |
… |
… |
Conversion rate (%) |
- |
- |
- |
… |
… |
… |
Table 4
Number of decimals in case of mean value and standard deviation must be equal.
p-value must be used (sample from population), with 3 decimals in each row of table.
Page 13, second line
‘… are belong to GH1;…’ font mut be adapted to the rest of text.
Page 13, second paragraph (‘…In soybeans and most soybean-derived foods…’)
Line spacing in first three lines of paragraph must be adapted to the rest of the text.
References subsection
Only in case of doi on the end first part of link https:// is required
https://doi.org/10.1002/mnfr.20200001 in case of reference no.1.
Reviewer 2 Report
Comments and Suggestions for Authors
It is a very interesting document. It raises the generation of scientific knowledge that can be of great use to science. There are only a few minor flaws. The abstract should have a better structure. The introduction does not address important aspects such as further justifying the rationale for evaluating the Clostridium strain. Addressing the alternative use of β-glucosidases as a replacement for antibiotics, which is a worldwide concern. The methodology needs to integrate aspects on the ethical handling of animals during experimentation and evidence of a protocol that ensures animal welfare. It is necessary to indicate how the birds were housed, spaces, areas used, environmental conditions and feed management, and to add more specific aspects of the strain used as inoculum. In the results, the graphs are not very clear. They should be larger and clearer for visual clarity and interpretation. The discussion should be reinforced by comparison with previous studies where β-glucosidases have been used and their effect on immunity.

Round 2
Reviewer 2 Report
Comments and Suggestions for Authors
I would have liked to see a corrected version (clean version) of the document. The current version complicates a proper reading of the paper. The abstract was structured in a more appropriate way as requested by the Journal. The introduction was improved and enriched with information related to how genistein stimulates immunity. In addition, relevant information on Clostridium perfringens in poultry was included. The methodology was also complemented and restructured with a more reader-friendly scheme. In the results section, the suggested changes have been made, but the graphs are still difficult to understand due to the size and sharpness of some of them.
